# Zika Virus Antibody Titers Three Years after Confirmed Infection

**DOI:** 10.3390/v13071345

**Published:** 2021-07-12

**Authors:** Thomas Langerak, Louella M. R. Kasbergen, Felicity Chandler, Tom Brinkman, Zéfia Faerber, Kajal Phalai, Sebastian Ulbert, Alexandra Rockstroh, Erwin de Bruin, Marion P. G. Koopmans, Barry Rockx, Eric C. M. van Gorp, Stephen Vreden

**Affiliations:** 1Erasmus Medical Center, Department of Viroscience, 3015 GD Rotterdam, The Netherlands; thomas.langerak@erasmusmc.nl (T.L.); l.kasbergen@erasmusmc.nl (L.M.R.K.); f.chandler@erasmusmc.nl (F.C.); brinkman.tk@gmail.com (T.B.); e.debruin@erasmusmc.nl (E.d.B.); m.koopmans@erasmusmc.nl (M.P.G.K.); b.rockx@erasmusmc.nl (B.R.); e.vangorp@erasmusmc.nl (E.C.M.v.G.); 2Department of Internal Medicine, Academic Hospital Paramaribo, Paramaribo, Suriname; zefiamariah@live.com (Z.F.); phalai_kaj@hotmail.com (K.P.); 3Fraunhofer Institute for Cell Therapy and Immunology, 04103 Leipzig, Germany; sebastian.ulbert@izi.fraunhofer.de (S.U.); alexandra.rockstroh@izi.fraunhofer.de (A.R.)

**Keywords:** Zika virus, serology, cross-reactivity, dengue virus, waning immunity

## Abstract

Background: In 2015–2016, a large Zika virus (ZIKV) outbreak occurred in the Americas. Although the exact ZIKV antibody kinetics after infection are unknown, recent evidence indicates the rapid waning of ZIKV antibodies in humans. Therefore, we aimed to determine the levels of ZIKV antibodies more than three years after a ZIKV infection. Methods: We performed ZIKV virus neutralization tests (VNT) and a commercial ZIKV non-structural protein 1 (NS1) IgG ELISA in a cohort of 49 participants from Suriname who had a polymerase-chain-reaction-confirmed ZIKV infection more than three years ago. Furthermore, we determined the presence of antibodies against multiple dengue virus (DENV) antigens. Results: The ZIKV seroprevalence in this cohort, assessed with ZIKV VNT and ZIKV NS1 IgG ELISA, was 59.2% and 63.3%, respectively. There was, however, no correlation between these two tests. Furthermore, we did not find evidence of a potential negative influence of DENV immunity on ZIKV antibody titers. Conclusions: ZIKV seroprevalence, assessed with two commonly used serological tests, was lower than expected in this cohort of participants who had a confirmed previous ZIKV infection. This can have implications for future ZIKV seroprevalence studies and possibly for the duration of immunological protection after a ZIKV infection.

## 1. Introduction

Zika virus (ZIKV) is an arthropod-borne virus within the Flaviviridae family that was first identified in Uganda in 1947 and has since then circulated largely unnoticed in Africa and Asia [1,2,3]. In 2015–2016, a large ZIKV outbreak occurred in the Americas, leading to several hundreds of thousands of confirmed ZIKV infections, although due to the high number of asymptomatic infections, estimates of the total count of infections during the outbreak are in the order of hundreds of millions [4]. During this outbreak, previously unreported complications of a ZIKV infection were observed: the Guillain–Barré syndrome and congenital abnormalities in the offspring of mothers who were infected with ZIKV during pregnancy [5,6].

Currently, little to no ZIKV circulation is reported in the countries that were affected during the 2015–2016 ZIKV outbreak. One of the reasons for this might be herd immunity; seroprevalence studies performed relatively soon after the 2015–2016 ZIKV outbreak in the Americas reported a ZIKV seroprevalence between 20–60% in the affected countries [7,8,9,10]. However, two recently published studies reported rapid waning of ZIKV antibodies [11,12].

Antibody cross-reactivity between flaviviruses is a well-known problem that makes flavivirus serology extremely challenging. Virus neutralization tests (VNTs), in which nAbs are detected, are considered to be the gold standard for flavivirus serology. We and others have previously shown that VNTs are well suited to distinguish dengue virus (DENV) and ZIKV nAbs and to determine that the cross-neutralization of ZIKV by DENV antibodies does not occur regularly, especially not in non-acute sera [7,13,14]. On the contrary, we demonstrated that for the ZIKV NS1 IgG ELISA, antibody cross-reactivity between other flaviviruses and ZIKV is a significant problem [7]. The cut-off for a positive ZIKV VNT result is often based on sera from patients who had a recent ZIKV infection that was compared with sera from patients who had a previous infection or vaccination with other flaviviruses such as DENV, yellow-fever virus (YFV), tick-borne encephalitis virus (TBEV), West Nile virus (WNV) and Japanese encephalitis virus (JEV). However, in the case of waning ZIKV nAb titers, it could be that the ZIKV VNT is less well-suited to use as a serological test to detect ZIKV exposure several years after a ZIKV infection because of a loss of sensitivity. This potential loss of ZIKV VNT sensitivity several years after an infection can have important implications for future ZIKV seroprevalence studies because of the possible underestimation of previous ZIKV spread.

Here, we assessed the ZIKV nAb titers and antibodies against ZIKV NS1 in sera from 49 participants in Suriname who had a reverse transcriptase polymerase chain reaction (RT-PCR)-confirmed ZIKV infection more than three years ago. Furthermore, we determined nAbs against DENV-2 and binding antibodies against multiple DENV antigens in order to assess the effect of previous DENV exposure(s) on ZIKV antibody titers.

## 2. Materials and Methods

### 2.1. Study Population

People living in Suriname and aged 18 years or older who previously had a symptomatic, RT-PCR-confirmed ZIKV infection were asked by study personnel to participate in this study. Participants were asked to fill out a questionnaire about their health, YFV vaccination history and their pregnancy status during their ZIKV infection.

### 2.2. Ethical Approval

Approval for this study was granted by the national medical ethical board of Suriname. Written informed consent was obtained from all participants. This study was performed according to the principles of the Declaration of Helsinki.

### 2.3. Serology

One tube of blood was collected via venipuncture for serum isolation. Serum was stored at −20 °C until shipment to the Netherlands for analysis. All serological analyses were performed at the World Health Organization Collaborating Centre for Arbovirus and Haemorrhagic Fever Reference and Research at Erasmus Medical Centre in Rotterdam, the Netherlands. IgG antibodies against ZIKV NS1 were assessed using a commercial ELISA kit according to the manufacturers’ instructions (Euroimmun, Lubeck, Germany). The recommended cut-offs for this test are an ELISA ratio >1.1 for a positive result, an ELISA ratio between 0.8–1.1 for an equivocal result and <0.8 for a negative test result. A protein microarray was performed as previously described in detail with few exceptions, to detect IgG antibodies against several DENV and ZIKV antigens [15,16]. Slides were printed with DENV1–4 and ZIKV NS1 proteins (Sino Biological Europe GmbH and Immune Technology Corp., New York, NY) and Equad proteins (DENV envelope proteins containing four amino acid mutations in the highly conserved fusion loop domain to reduce flavivirus cross-reactivity) [17]. Slides were incubated with four-fold serially diluted sera ranging from 1:20 to 1:20,480. IgG binding was detected by incubation with Alexa Fluor 647 conjugated goat anti-human IgG-Fcγ (Jackson Immunoresearch), and signals were measured using the Tecan PowerScanner at 647 nm. The median fluorescent intensity of individual spots was used to calculate the IgG titer values at which the fluorescent curve crosses 50 percent of the maximum fluorescent value. Calculations were done using R studio software. Since this assay is currently not used for routine diagnostics, a cut-off for a positive result has not yet been determined.

### 2.4. Virus Neutralization Tests

The presence of ZIKV- and DENV-2-specific neutralizing antibodies was determined with an in-house virus neutralization test as described before [7,18]. In short, 100 TCID_50_ of dengue-2 (16,681 strain [19]) or an Asian lineage ZIKV (Suriname strain 2016, GenBank reference KU937936) was incubated with two-fold serial diluted serum and transferred to a confluent monolayer of Vero cells for one hour at 37 °C and 5% CO_2_. Subsequently, cells were washed three times and incubated with Dulbecco’s modified Eagle’s medium supplemented with 10% fetal bovine serum for five days for ZIKV and seven days for DENV-2. Readout was performed by the detection of a cytopathic effect with a light microscope. All samples were tested in triplicate and the geometric mean of the highest final serum dilution that completely prevented the cytopathic effect was calculated for all samples. The cut-off for a positive ZIKV and DENV-2 VNT result was set at a final serum dilution above 1:32 based on an in-house validation process in which ZIKV and DENV-2 convalescent sera were compared with each other and, for ZIKV VNT, with sera from people who had a previous infection or vaccination with other flaviviruses (YFV, TBEV, WNV and JEV).

### 2.5. Statistical Analyses

Statistical differences between sex, comorbidities and age within the ZIKV VNT or ZIKV NS1 IgG ELISA positive/negative groups were assessed with Pearson χ^2^ test and a t-test. Differences in ZIKV and DENV-2 nAb titers were tested with the Mann–Whitney *U* test. Spearman correlation was used to test for correlations between the different serological tests used in this study. All statistical analyses were performed with IBM SPSS for Windows, version 24. A *P* value ≤ 0.05 was considered to be a statistically significant difference.

## 3. Results

In total, 49 participants were recruited for this study, of which 39 were females (79.6%) and 10 were males (20.4%). The average age of the participants was 41.3 years (Table 1). The average time between the initial RT-PCR confirmed symptomatic ZIKV infection and the blood collection for this study was 3.2 years (range 3.1–3.5 years). No considerable differences were found between ZIKV-seropositive and -seronegative participants regarding sex, age and pregnancy status during ZIKV infection (Table 1). Yellow-fever-virus vaccination status did not differ between the ZIKV-seropositive and -seronegative groups. Comorbidities such as cardiovascular disease were more common in participants with a positive ZIKV VNT compared to participants with a negative ZIKV VNT (80% versus 20%, respectively, *P* = 0.05, Table 1).

Based on the ZIKV VNT results, the ZIKV seroprevalence in this cohort of participants who had a RT-PCR-confirmed ZIKV infection in the past was 59.2% (95% CI 44.2–73.0, Figure 1A). Based on the ZIKV NS1 IgG ELISA, ZIKV seroprevalence was 63.3% (95% CI 48.3–76.6, Figure 1A). Even though the ZIKV VNT and the NS1 IgG ELISA had comparable results regarding ZIKV seroprevalence, there was no correlation between ZIKV nAb titers and the ratios from the ZIKV NS1 IgG ELISA (r = 0.04, *P* = 0.80, Figure 2B). Additionally, of the 31 participants with a positive ZIKV NS1 IgG ELISA result, only 18 participants (58.1%) had a positive ZIKV VNT test result, further indicating the poor correlation between these tests. We subsequently performed VNTs for DENV-2 and found that DENV-2 seroprevalence in this cohort was 73.5% (95% CI 58,985.1). DENV-2 nAb titers were higher than ZIKV nAb titers (median titer 40 versus 64 *P* = 0.03, Figure 1C). Interestingly, five participants (10.2%, 95% CI 3.4–22.2) had no detectable ZIKV nAbs (Figure 1C). Antibody cross-reactivity, and to a lesser extent cross-neutralization, between ZIKV and DENV has been extensively reported [7,13,20]. We did not find a correlation between the DENV-2 nAb titers and the ZIKV nAb titers, indicating that cross-neutralization did not seem to occur in our assay with these samples (r = −0.07, *P* = 0.63, Figure 1D). Contrary to ZIKV nAb titers, there was a moderate positive correlation between ZIKV NS1 IgG ELISA ratios and DENV-2 nAb titers (r = 0.58, *P* < 0.001, Figure 1E). This might indicate the detection of cross-reactive antibodies towards DENV using the ZIKV NS1 IgG ELISA.

DENV and ZIKV are closely related, and it has been speculated that DENV exposure might negatively influence mounting a long-lasting ZIKV immune response upon ZIKV exposure [11]. Therefore, we tried to assess the role of DENV immunity on ZIKV nAb titers by performing a protein microarray (PMA) to measure IgG antibodies against DENV1–4 Envelope protein (E) and NS1 antigens and ZIKV NS1 IgG antibodies measured by the protein microarray. ZIKV and DENV-2 nAb titers and ZIKV NS1 IgG ELISA ratios are displayed in the heatmap in Figure 2. Very high antibody titers for all DENV E and NS1 antigens were observed in sera from participants with high ZIKV nAb titers (e.g., participants 45 and 21, Figure 2A). However, we also observed high antibody titers against DENV in sera from participants with low ZIKV nAb titers (e.g., participants 3, 18, 35, 47 and 12, Figure 2A). ZIKV NS1 IgG ELISA ratios correlated, in general, relatively well with DENV NS1 antibody titers determined with the PMA (e.g., in participants 12, 45 and 3, Figure 2B). As expected, ZIKV NS1 PMA titers strongly correlated with ZIKV NS1 IgG ELISA ratios (r = 0.91, *P* < 0.0001, Appendix AA). As seen for the ZIKV NS1 ELISA ratios, the ZIKV NS1 PMA titers correlated well with DENV-2 nAb titers (r = 0.48, *P* < 0.001, Appendix AB) but not with ZIKV nAb titers (r = 0.01, *P* = 0.55, Appendix AC).

## 4. Discussion

In this study, we found a ZIKV seroprevalence of 59.2% determined by VNT and 63.3% determined by ZIKV NS1 IgG ELISA in a cohort of 49 participants who had a RT-PCR-confirmed and symptomatic ZIKV infection more than three years ago. Even though the ZIKV seroprevalence found with both tests was similar, there was no correlation between ZIKV nAb titers and ZIKV NS1 IgG ELISA ratios. This may be explained by detecting different antibody populations between these two tests, since NS1 antibodies are non-neutralizing. There was, however, a correlation between DENV-2 nAb titers and ZIKV NS1 ELISA ratios, indicating that this test might detect cross-reactive DENV NS1 IgG antibodies. The low ZIKV seroprevalence three years after infection is in line with previous studies, in which it was shown that nAbs and NS1 antibodies against ZIKV decline rapidly, and indicate that ZIKV serology in flavivirus endemic populations is very challenging several years after the initial infection [11,12]. This could possibly in part explain why the reported ZIKV seroprevalence in regions in Africa and Asia, where ZIKV has already been circulating for many years, is low [21,22,23]. One possible solution to increase the sensitivity of ZIKV serological assays in populations that did not recently suffer from a ZIKV outbreak is to set a lower cut-off for a positive test result. However, this will undoubtedly come at the cost of lower specificity due to the notorious antibody cross-reactivity between flaviviruses. Since a low degree of T cell cross-reactivity between flaviviruses has been reported, cellular assays might be useful to detect previous ZIKV infections in flavivirus endemic populations [24,25]. A limitation of this study is that its cross-sectional design makes it impossible to confirm whether the observed low ZIKV antibody titers in some participants are indeed due to waning, or if these participants never developed (high titer) antibodies against ZIKV. However, since it has been demonstrated that seroconversion after a ZIKV infection generally occurs, a lack of seroconversion is not likely to be the explanation for the low seroprevalence found in this cohort [26,27].

DENV-2 seroprevalence and DENV-2 nAb titers were higher compared to ZIKV seroprevalence and nAb titers, possibly due to multiple previous DENV exposures. To try to understand why some participants had low or absent antibodies against ZIKV, we determined binding antibodies against different DENV antigens with a protein microarray. It is assumed that a heterotypic secondary DENV infection often leads to the reactivation of memory B- and T-cells that were induced during the original previous DENV infection(s) via a mechanism called original antigenic sin (OAS). Because of OAS, DENV pre-immunity may negatively impact the establishment of a ZIKV-specific immune response [28,29]. We indeed found that some participants with high DENV antibody titers had low ZIKV nAb titers. However, we also found high ZIKV nAb titers in sera from participants with high DENV antibody titers. Therefore, we cannot conclude that OAS between DENV and ZIKV might play a role in establishing a long-lasting ZIKV immune response. However, there were several limitations with this analysis. Firstly, we looked at DENV antibodies three years after the participants had a ZIKV infection, which might not correlate to the situation at the time of ZIKV infection regarding (the amount of) DENV exposure. Secondly, flavivirus antibodies are notorious for cross-reactivity; this makes it very challenging to demonstrate an inverse correlation between the breadth and magnitude of ZIKV antibodies compared to DENV antibodies.

It is commonly assumed that a ZIKV infection results in long-term and possibly lifelong immunity against ZIKV; however, sufficient data to support this is currently lacking. Furthermore, for DENV serotypes 1 to 4, which are closely related to ZIKV, homotypic reinfections have been sporadically observed [30]. We show that in sera from five participants (10.2%), no ZIKV neutralization was observed with the highest serum concentration tested (titer < 1:8). However, since the minimum nAb titer required for protection against (re-)infection with ZIKV remains unknown and we did not look at the presence of memory B- or T-cells in this study, we were not able to determine whether individuals with low nAbs are possibly susceptible to ZIKV reinfection. Although the questions of the duration of ZIKV immunity remain unanswered, it is plausible that the absence of circulating ZIKV nAbs results in the loss of sterilizing ZIKV immunity. The loss of sterilizing ZIKV immunity could result in transient viremia upon secondary ZIKV exposure, until the effector cells of the adaptive immune system are reactivated and clear the virus. This potential short period of viremia upon secondary ZIKV exposure in persons with low or absent ZIKV nAbs can be of special importance to pregnant women, in whom ZIKV, during viremia, can possibly cross the placenta and infect the fetus. Furthermore, if the protection against ZIKV reinfection is shorter than expected, this can have important implications for when a new ZIKV outbreak can be expected in the regions that were affected during the 2015–2016 outbreak [31].

## 5. Conclusions

In conclusion, we found that ZIKV seroprevalence was relatively low in this cohort of participants with a previous ZIKV infection. These results indicate that caution is warranted in the interpretation of ZIKV seroprevalence data several years after a ZIKV infection. In order to gain more knowledge on ZIKV antibody dynamics and duration of protection after a ZIKV infection, longitudinal studies need to be performed that, preferably, also use cellular assays to determine previous ZIKV exposure.

## Figures and Tables

**Figure 1 viruses-13-01345-f001:**
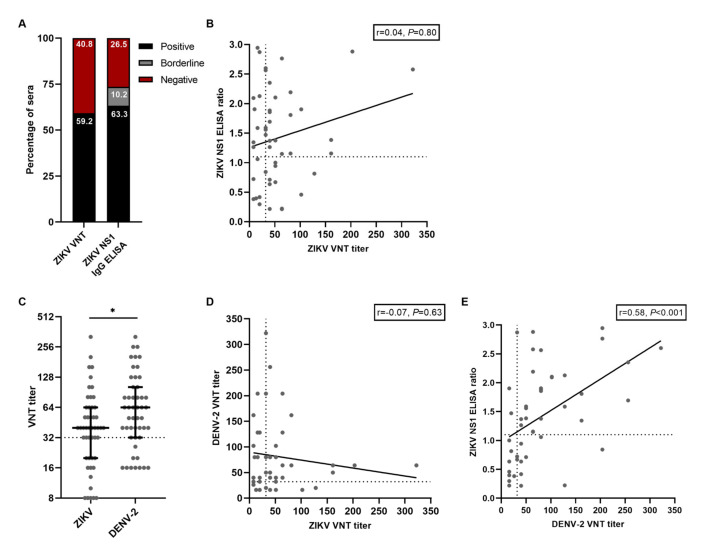
**Results from serological assays for ZIKV and DENV antibodies**. **A**: Percentage of positive and negative tested sera with ZIKV VNT and ZIKV NS1 IgG ELISA. **B**: Correlation between ZIKV NS1 IgG ELISA ratios and titers from the ZIKV VNT. The dotted lines indicate cut-off values for a positive test result. **C**: ZIKV- and DENV-2 VNT titers from all participants. Lines represent median ± IQR. The dotted line indicates the cut-off value for a positive test result. Statistical differences were tested with the Mann–Whitney test. **D**: Correlation between DENV-2 VNT titers and ZIKV VNT titers. The dotted lines indicate cut-off values for a positive test result. **E**: Correlation between ZIKV NS1 IgG ELISA ratios and DENV-2 VNT titers. The dotted lines indicate cut-off values for a positive test result. * *P* < 0.05.

**Figure 2 viruses-13-01345-f002:**
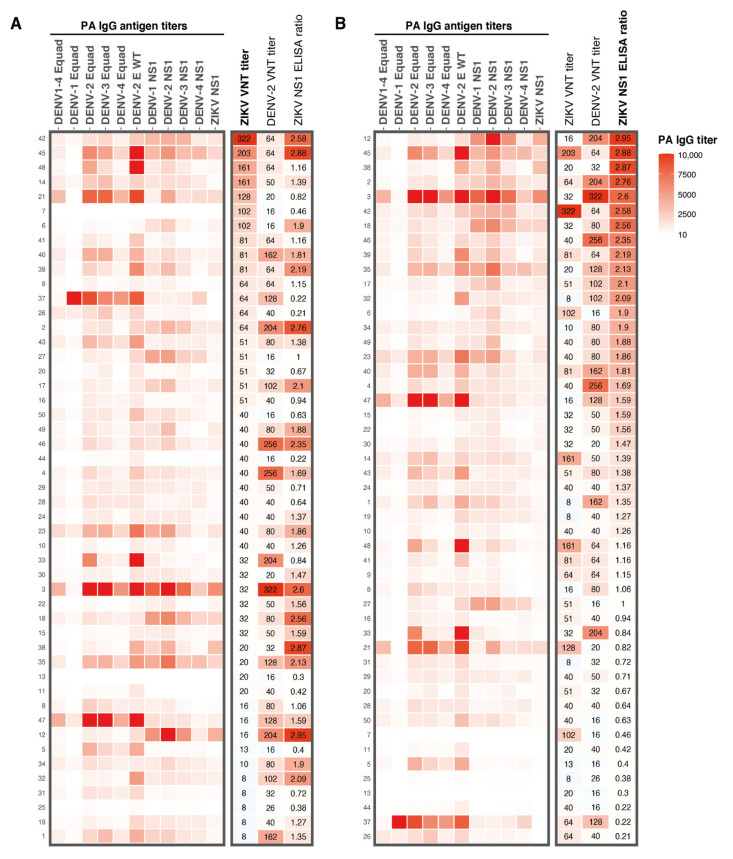
**Heatmap of results from the different serological assays used in this study**. **A**: IgG antibody titers for DENV1–4 Equad and DENV1–4 and ZIKV NS1 antigens determined with a protein microarray. Corresponding ZIKV and DENV-2 VNT titers and ZIKV NS1 IgG ELISA ratios from all participants are shown on the right. Antibody patterns are ranked from highest to lowest ZIKV VNT titer. **B**: Protein microarray IgG antibody titer patterns for DENV1–4 Equad and DENV1–4 and ZIKV NS1, ZIKV and DENV-2 VNT titers and ZIKV NS1 IgG ELISA ratios from all participants, ranked from highest to lowest ZIKV NS1 ELISA ratio. Numbers on the left Y-axis are the study numbers of the participants in this study. PA; protein microarray, Equad; envelope proteins containing four amino acid mutations in the highly conserved fusion loop domain to reduce flavivirus cross-reactivity, VNT; virus neutralization test.

**Table 1 viruses-13-01345-t001:** Baseline characteristics of the total cohort and Zika and dengue-2 VNT positive/negative participants.

	Total, *N* = 49	ZIKV VNT Pos. ^1^ *N* = 29	ZIKV VNT Neg. ^1^ *N* = 20	*P*	ZIKV NS1 ELISA IgG Pos. ^2^ *N* = 31	ZIKV NS1 ELISA IgG Borderline/Neg. ^2^ *N* = 18	*P*
Sex, *n* (%)							
Female	39 (79.6)	23 (59.0)	16 (41.0)	0.95	26 (66.7)	13 (33.3)	0.32
Age, mean (range)	41.3 (25–59)	39.8 (25–59)	43.4 (31–58)	0.20	41.3 (25–59)	41.2 (28–58)	0.97
Comorbidities ^3^, *n* (%)							
Yes	15 (30.6)	12 (80.0)	3 (20.0)	0.05	11 (73.3)	4 (26.7)	0.33
Yellow-fever-virus vaccinated, *n* (%)							
Yes	39 (79.6)	21 (53.8)	18 (46.2)		24 (61.5)	15 (38.5)	
Unknown	5 (9.8)	4 (80.0)	1 (20.0)		4 (80.0)	1 (20.0)	
No	5 (9.8)	4 (80.0)	1 (20.0)		3 (60.0)	2 (40.0)	
Pregnant during ZIKV infection, *n* (%)							
Yes	12 (24.5)	8 (66.7)	4 (33.3)		6 (50.0)	6 (50.0)	
Pregnancy with complications ^4^	2 (16.7)	0	2 (100)		0	2 (100)	

VNT; virus neutralization test, NS1; non-structural protein 1. ^1^ Cut-off for positive VNT result: >1:32. ^2^ Cut-off for ZIKV NS1 ELISA: <0.8 = negative, 0.8–1.1 = borderline, >1.1 = positive. ^3^ Comorbidities include cardiovascular diseases such as hypertension and diabetes mellitus and autoimmune diseases such as rheumatoid arthritis. ^4^ Pregnancy complications include spontaneous abortion, stillbirth and congenital abnormalities.

## Data Availability

Please refer to suggested Data Availability Statements in section “MDPI Research Data Policies” at https://www.mdpi.com/ethics.

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
