# Peer review of "Zika Virus Antibody Titers Three Years after Confirmed Infection"

_viruses, 2021, doi:10.3390/v13071345_

Round 1
Reviewer 1 Report
This paper is a study of serological changes after 3 years in 49 ZIKV-infected patients. However, it is unfortunate that there is no data such as neutralizing antibody values at the time of initial infection (3 years ago).
The ZIKV seroprevalence in this cohort, assessed with ZIKV VNT and ZIKV NS1 IgG ELISA, was 59.2% and 63.3%, respectively.
45, 42, 48, 14, 21, 7, and 6 showed neutralizing antibody titers greater than 100.
However, a logical explanation is needed as to why it is likely to maintain such high neutralizing antibody titers.
In particular, 21,7 showed no correlation due to the low NS1 ELISA antibody titer, and this needs explanation.
Author Response
We would like to thank the reviewer for reading the manuscript and providing us with comments. We agree that it is unfortunate that we do not have sera from when the patients got diagnosed with ZIKV (or short after) and therefore we do not know the ZIKV antibody titers at the time of infection. We added this information as a limitation of this study in line 235-240.
Regarding the following comment from the reviewer:
‘45, 42, 48, 14, 21, 7, and 6 showed neutralizing antibody titers greater than 100. However, a logical explanation is needed as to why it is likely to maintain such high neutralizing antibody titers.’
Even though our main observation is that many participants have low or absent ZIKV neutralizing antibody titers, it is indeed notable that some participants have rather high neutralizing antibody titers. Since this study is cross-sectional, we can unfortunately only speculate why this is the case. It could be that these participants were recently exposed or vaccinated against other flaviviruses and that cross-neutralization of ZIKV is observed since cross-neutralization of ZIKV has been described in the early phases after a DENV infection (M. Montoya et. al, JID, 2018, DOI: 10.1093/infdis/jiy164). However, this hypothesis is not very likely since titers of DENV-2 neutralizing antibodies are low. It could also be that these participants have been exposed to ZIKV since their ZIKV infection ~3 years ago which can boost the antibody titers without giving a symptomatic infection, as has recently been shown for DENV (L.W. Alexander, PNAS, 2021, DOI: 10.1073/pnas.2013941118). However, there has been little to no reported ZIKV circulation in South America since 2017. Another, and possibly most likely, option is that this observation is part of the normal distribution of antibody titers in a population after a (viral) infection. This has also be observed by Henderson et. al (eLife, 2020, DOI: 10.7554/eLife.48460) in which ZIKV PRNT titers of several donors are far above 100 1-2 years after a (potential) ZIKV infection (Figure S4).
Regarding the comment: ‘In particular, 21,7 showed no correlation due to the low NS1 ELISA antibody titer, and this needs explanation.’
In participant 21 and 7, but also in several other participants, there is indeed no correlation at all between the ZIKV neutralizing antibody titers and the ZIKV NS-1 IgG titers. In the manuscript we mention this lack of correlation between the results of these assays multiple times (line 19-20, line 144-146, line 201-203 and Figure 1B). There was, however, a correlation between DENV-2 nAb titers and ZIKV NS1 ELISA ratios indicating that this test might detect cross-reactive DENV NS1 IgG antibodies which can explain the lack of correlation between ZIKV NS1 IgG ELISA and ZIKV VNT. Furthermore, another possible explanation for this is that both assays detect different types of antibodies and that NS1 antibodies are non-neutralizing, we added this to the manuscript (line 204-205).
Reviewer 2 Report
Here the authors do serology on ZIKV infected patients 3 years after confirmed infection with and recovery from ZIKV. The goal is to determine if there remains significant antibodies to ZIKV or is there deterioration of the humoral immune response over time. The authors use Viral Neutralization Tests with titration of sera and also use NS1 ELISA tests to determine antibody levels. The results indicate that ~3 years after ZIKV infection approximately 59% are VNT positive and 63% are NS1 ELISA positive. The authors indicate that this is evidence of humoral immune response deterioration over time. This is a very concise clinical serology paper with much needed piece of information on humoral immunity to ZIKV in the long term. There are several concerns.
- Do the authors have data on antibody titers to ZIKV immediately after the infection in 2015-2016? Since they propose that the antibody response is fading to the previous ZIKV infection, perhaps the percentage of seropositive during the immediate convalescence is not significantly different that the data here. Please provide this immediate convalescence serology if it is available.
- In the two previous studies on waning ZikV antibodies, how does that data compare with the waning antibodies here?
Author Response
We would like to thank the reviewer for reading the manuscript and providing us with comments.
- Regarding the comment: ‘Do the authors have data on antibody titers to ZIKV immediately after the infection in 2015-2016?’
Unfortunately we do not have sera from these patients during or short after their ZIKV infection and therefore we cannot test the initial antibody titers that these participants had. We clarified this in the manuscript (line 235-238).
- Regarding the comment: ‘Since they propose that the antibody response is fading to the previous ZIKV infection, perhaps the percentage of seropositive during the immediate convalescence is not significantly different that the data here.’ Since we do not have the initial titers, it could indeed be that some participants never seroconverted and that this explains our observations. However, from what is known from literature, absence of seroconversion after a ZIKV infection is described but is a rare phenomenon. We therefore think that this cannot solely explain the low seroprevalence in this cohort. We did add a few lines of discussion on this topic to the manuscript (line 235-240)
- Regarding the comment: ‘In the two previous studies on waning ZikV antibodies, how does that data compare with the waning antibodies here?’
A direct comparison between the two previous studies on waning ZIKV antibodies and this study is difficult due to the following reasons:
- The two other studies are longitudinal studies whereas our study is cross-sectional, therefore we cannot call the observed low antibody titers in our study ‘waning’.
- Our cohort consists of participants with a previously confirmed ZIKV infection whereas this is not the case in the two other studies.
- Our study is performed >3 years after the initial infection while the other two studies are performed 1-2 years after the peak of the 2015-2016 ZIKV epidemic.
- For the study of Henderson et. al; different assays were used in this study compared to our study which makes it difficult to compare the titers and seroprevalence.
We can compare the median ZIKV NS1 ELISA ratios from the participants in our cohort with the participants from the study of Moreira-Soto et. al (DOI: 10.1007/s11262-020-01772-2) that were initially ZIKV seropositive. Moreira-Soto et. al report that for participants that were initially ZIKV seropositive, 1.5-2 years later the median ZIKV NS1 ELISA ratios were 1.6 while in our cohort median ZIKV NS1 ELISA ratios were 1.37.
Other changes:
- Throughout the manuscript and in the title we changed ‘anti-ZIKV antibodies’ and ‘anti-DENV antibodies’ in ‘ZIKV antibodies’ and ‘DENV antibodies’.
- We added the initials of two authors.
Round 2
Reviewer 1 Report
The revision seems to have satisfied the question.
Author Response
We thank reviewer 1 for his/her efforts and comments. Since there were no new comments in round 2, we did not make additional changes to the manuscript.